# Developmental Changes of Immunity and Different Responses to Weaning Stress of Chinese Indigenous Piglets and Duroc Piglets during Suckling and Weaning Periods

**DOI:** 10.3390/ijms232415781

**Published:** 2022-12-12

**Authors:** Sujuan Ding, Yating Cheng, Md. Abul Kalam Azad, Qian Zhu, Pan Huang, Xiangfeng Kong

**Affiliations:** 1Key Laboratory of Agro-Ecological Processes in Subtropical Region, Hunan Provincial Key Laboratory of Animal Nutritional Physiology and Metabolic Process, Institute of Subtropical Agriculture, Chinese Academy of Sciences, Changsha 410125, China; 2College of Advanced Agricultural Sciences, University of Chinese Academy of Sciences, Beijing 100008, China

**Keywords:** Duroc pig, immune function, Taoyuan black pig, weaning stress, Xiangcun black pig

## Abstract

To investigate developmental changes in immunity and different responses to weaning stress of piglets from different breeds during suckling and weaning periods, a total of 30 litters of Taoyuan black (TB) piglets, Xiangcun black (XB) piglets, and Duroc (DR) piglets (ten litters per breed) were selected at 1, 10, 21, and 24 days of age, respectively. The results showed that the liver index of TB piglets was higher at 10 days of age than that of the other days of age and breeds. Regardless of the days of age, TB and XB piglets had a higher plasma IgA level and lower ileal IgM level than in the DR piglets, and XB piglets had a lower plasma IgG level than the other breeds. TB and XB piglets had a higher IL-6 level and lower IL-17 level in plasma at 24 days of age than DR piglets, regardless of the days of age. The ileal levels of IL-2, IL-10, IFN-γ, and TNF-α were lower in the TB and XB piglets at 24 days of age than in the DR piglets. The ileal expression levels of *IRAK1*, *CD14*, *MyD88*, and *NF-κB* were down-regulated in the TB and XB piglets at 24 days of age compared to those in the DR piglets. These findings suggest that there were differences in the development of immune function among different pig breeds. Moreover, TB and XB piglets presented stronger resistance to weaning stress than the DR piglets, which may be related to the immune regulation mediated by the *MyD88*/*NF-κB* signaling pathway.

## 1. Introduction

The development of the immune system of pigs during the pre- and post-natal periods is one of major importance for a healthy pig. The gastrointestinal tract is one of the largest immune organs in pigs, and the intestinal immune barrier function is the first line of defense against invading pathogens [1]. The intestinal mucosal lamina propria consists of T and B cells at birth, and the number of these cells generally increases two-fold at four weeks after birth [2]. During the lactation period, breast milk provides not only the nutrients needed for growth and development but also an important source of several antibodies for the passive immune function of piglets [3]. For example, immunoglobulin A (IgA) in breast milk is one of the most important protective antibodies and plays a crucial role in the development of the intestinal tract of piglets. After birth, the piglets can obtain antibodies such as IgA and IgG from the colostrum of sows, and these antibodies can be transported into the newborn piglets for efficient development and immune reactions [4]. Therefore, understanding the developmental changes in the immune profiles of piglets is critical for the growth and development of healthy pigs.

Early weaning is the biggest stress factor for piglets after birth, accompanied by changes in intestinal morphology, microbiota, and immune status, leading to diarrhea and affecting the growth and development of piglets [5]. At this stage, the intestine is highly influenced by a potential interaction between nutrition, the intestinal mucosal membrane, and the intestinal microbiota, which affect the intestinal physiology, health, and well-being of the piglet. The mucosal immune system has two functions: generating an active immune response to pathogens and developing immune tolerance to innocuous food and commensal antigens [6]. Birth and weaning are the most important periods of exposure to neoantigens for young animals [7]. Compared with neonatal piglets, weaned piglets lose the immune protection of breast milk, and the mucosal immune system produces an active immune response [8]. Furthermore, weaning also leads to the impaired immune function of the small intestine of piglets, resulting in a decline in lymphocyte immune function, increases in the intestinal pro- and anti-inflammatory cytokines and related gene expressions, and changes in intestinal morphology [9,10]. These changes can adversely influence digestion and absorption of nutrients, secretion of antibodies, and maintenance of the intestinal barrier function, thereby reducing the growth performance of post-weaning pigs [11]. Hence, it is necessary to reveal the changes in immune function during weaning to prevent weaning stress.

Chinese indigenous pigs have shown good adaptability to the specific feeding environment and feed resources in intensive production and have the characteristics of resistance to adverse reactions (e.g., pre- and post-weaning stress, roughage feeding, etc.) [12]. Taoyuan black (TB) pigs, a typical indigenous pig breed in China, have the characteristics of roughage feeding tolerance, strong stress resistance, robust physique, and excellent meat quality. However, the growth rate of TB pigs is slower than foreign pig breeds [13]. Xiangcun black (XB) pigs, a cross-breed of TB pigs (as the female parent) and Duroc (DR) pigs (as the male parent), have the characteristics of strong stress resistance, large litter size, faster growth and development, high feed utilization rate, and excellent meat quality. The research on TB and XB pigs has been limited to growth performance, meat quality, and roughage tolerance; however, the research on the intestinal immune function of these pig breeds during suckling and weaning periods is still very limited. Therefore, we hypothesized that understanding the underlying mechanism of developmental changes in the intestinal immune function of indigenous pig breeds will shed new light on nutrition approaches to improve their health during suckling and weaning periods. Thus, TB, XB, and DR piglets during suckling (1, 10, and 21 days of age) and weaning (24 days of age) periods were selected from the same litter of each breed to investigate developmental changes in immunity and different responses to weaning stress during these periods.

## 2. Results

### 2.1. Changes in Liver and Spleen Indexes of Different Pig Breeds during Suckling and Weaning Periods

The changes in organ indexes of different pig breeds during suckling and weaning periods are presented in Table 1. The liver index of TB piglets was higher (*p* < 0.05) at 10 days of age than that at other days of age, as well as at 21 and 24 days of age compared to 1 day of age. Moreover, the liver index of TB piglets was higher (*p* < 0.05) than that of the XB and DR piglets at 10 days of age, and the liver index of XB piglets was higher (*p* < 0.05) than that of the DR piglets at 21 days of age. At 1 day of age, the spleen index of TB piglets was lower (*p* < 0.05) than that at other days of age, and the spleen index of XB piglets was lower (*p* < 0.05) than that at 10 and 21 days of age. In addition, the spleen index did not differ (*p* > 0.05) among the three pig breeds, regardless of days of age.

### 2.2. Changes in Hematological Parameters of Different Pig Breeds during Suckling and Weaning Periods

The changes in blood leukocyte composition of different pig breeds during suckling and weaning periods are shown in Figure 1. In TB piglets, the neutrophil count (Neu) level was higher (*p* < 0.05) at 21 days of age than at other days of age, while the absolute values of the lymphocyte (Lym) level and the lymphocyte ratio (Lym%) were higher (*p* < 0.05) at 24 days of age than those at other days of age. The white blood cell count (WBC) level of TB piglets was lower (*p* < 0.05) at 1 day of age than at other days of age, while it was also lower (*p* < 0.05) at 10 days of age than at 24 days of age. The neutrophil ratio (Neu%) of TB piglets was higher (*p* < 0.05) at 1 and 21 days of age than at 10 and 24 days of age. Among different pig breeds at the same age, the Neu and basophil count (Bas) levels and Neu% at 21 days of age and basophil ratio (Bas%) at 24 days of age were higher (*p* < 0.05) in the TB piglets compared with those in the XB and DR piglets. The Neu and Lym levels, Neu%, and monocyte ratio (Mon%) of XB piglets were lower (*p* < 0.05), but the Lym% was higher (*p* < 0.05) at 24 days of age than those in the TB and DR piglets. Moreover, the WBC and Neu levels at 1 day of age and Lym level at 1 and 21 days of age were lower (*p* < 0.05) in the TB and XB piglets compared with the DR piglets, and the Lym% at 21 days of age was lower (*p* < 0.05) in the TB piglets compared with the XB and DR piglets.

The changes in blood cell-related hematological parameters of different pig breeds during suckling and weaning periods are presented in Appendix A. In TB piglets, the hemoglobin (HGB) and hematocrit (HCT) levels were higher (*p* < 0.05) at 24 days of age than at 1, 10, and 21 days of age, whereas the coefficient of variation of red blood cell volume distribution width (RDW-CV) and standard deviation in red cell distribution width (RDW-SD) levels were lower (*p* < 0.05) at 1 and 24 days of age than those at 10 and 21 days of age. In XB piglets, the HGB and HCT levels were higher, while the mean platelet volume (MPV) level was lower at 24 days of age than at 1, 10, and 21 days of age (*p* < 0.05). In DR piglets, the mean corpuscular volume (MCV) and MPV levels were lower, while the mean red blood cell hemoglobin concentration (MCHC) level was higher at 24 days of age than at 1, 10, and 21 days of age (*p* < 0.05). Among different pig breeds at the same age, TB and XB piglets had lower (*p* < 0.05) red blood cell count (RBC), HGB, and HCT levels at 10 days of age, RBC and HCT levels at 21 days of age, and MCHC level at 24 days of age, whereas platelet count (PLT), platelet distribution width (PDW), and plateletcrit (PCT) levels at 10 days of age, mean red blood cell hemoglobin (MCH) level at 21 days of age, and MCV level at 24 days of age were higher (*p* < 0.05) compared with the DR piglets.

### 2.3. Changes in Plasma and Ileal Immunoglobulin Levels of Different Pig Breeds during Suckling and Weaning Periods

The changes in plasma immunoglobulin (Ig) levels of different pig breeds during suckling and weaning periods are presented in Table 2. The IgM level in TB piglets was lower (*p* < 0.05) at 21 and 24 days of age than at 1 day of age. However, there were no significant changes (*p* > 0.05) in IgG and IgA levels in TB piglets during different days of age. In XB piglets, the IgG level was higher (*p* < 0.05) at 24 days of age than at other days of age, and the IgA level was lower (*p* < 0.05) at 10 days of age than at 21 and 24 days of age, while there were no changes (*p* > 0.05) in IgA and IgM levels at different days of age. The IgA level of DR piglets was higher (*p* < 0.05) at 24 days of age than at other days of age, and the IgM level at 1 day of age was higher (*p* < 0.05) than at other days of age. There was no significant difference (*p* > 0.05) in the IgG level in DR piglets at different days of age. Among different pig breeds at the same age, the IgG level in the TB and DR piglets was higher (*p* < 0.05) during 1–24 days of age compared with the XB piglets, and the IgA level in the TB and XB piglets was higher (*p* < 0.05) during 1–21 days of age compared with the DR piglets. In addition, the IgA level in the XB piglets was higher (*p* < 0.05) at 24 days of age compared with the TB and DR piglets, and the IgM level in the XB piglets was lower (*p* < 0.05) at 1 day of age compared with the DR piglets.

The changes in ileal immunoglobulin levels of different pig breeds during suckling and weaning periods are listed in Table 3. In TB piglets, the secretory IgA (sIgA) level was higher (*p* < 0.05) at 21 and 24 days of age than at 1 and 10 days, as well as the IgG, IgA, and IgM levels at 10 days of age compared with those at 1 day of age. Moreover, the IgA and IgM levels at 21 days of age and IgG level at 24 days of age were higher (*p* < 0.05) compared with those at 1 and 10 days of age. In XB piglets, the IgA and IgG levels were lower (*p* < 0.05) at 1 day of age than at other days of age, as well as IgG level at 21 days of age than at 10 days of age; the IgM level was lower (*p* < 0.05) at 1 and 21 days of age than at 10 and 24 days of age, while the IgM level at 21 days of age was higher (*p* < 0.05) than at 1 day of age. Among different pig breeds at the same age, the ileal IgG level at 1 and 21 days of age and IgM level during 1–24 days of age were lower (*p* < 0.05) in the TB and XB piglets than those in the DR piglets, and the IgM level in the XB piglets was higher (*p* < 0.05) at 24 days of age compared with the TB piglets; the IgA level in the TB piglets was lower (*p* < 0.05) at 1 and 24 days of age than that in the DR piglets; the sIgA level in the TB and XB piglets was lower (*p* < 0.05) at 1 and 21 days of age than that in the DR piglets, and the sIgA level in the TB piglets was lower (*p* < 0.05) at 10 and 24 days of age than that in the XB and DR piglets. In addition, IgG level in the TB and XB piglets tended to be lower (*p* = 0.051) than that in the DR piglets at 24 days of age.

### 2.4. Changes in Plasma and Ileal Immuno-Cytokine Levels of Different Pig Breeds during Suckling and Weaning Periods

The changes in plasma immuno-cytokine levels of different pig breeds during suckling and weaning periods are presented in Figure 2. In TB piglets, the interferon (IFN)-γ level was higher (*p* < 0.05) at 1 and 10 days of age than at 21 and 24 days of age, as well as the tumor necrosis factor (TNF)-α level at 1 day of age compared with the other days of age. In XB piglets, the interleukin (IL)-6 level was lower (*p* < 0.05) at 1 and 10 days of age than at 21 and 24 days of age, while the TNF-α level was higher (*p* < 0.05) at 24 days of age than at 21 days of age; the TNF-α level was also higher (*p* < 0.05) at 1 day of age than at 10 and 21 days of age. In DR piglets, the IL-10 and IL-17 levels were higher (*p* < 0.05) at 1 day of age than at other days of age, and the IL-2 level was higher (*p* < 0.05) at 1 day of age than at 21 and 24 days of age. Among different pig breeds at the same age, the IFN-γ and TNF-α levels at 1 and 10 days of age were higher (*p* < 0.05) in the TB piglets than those in the XB and DR piglets. The IL-1β level at 1 day of age was lower in the TB and XB piglets than that in the DR piglets, whereas the IL-6 level at 21 days of age was higher in the XB piglets than that in the TB and DR piglets (*p* < 0.05). Compared with the DR piglets, TB and XB piglets had lower levels of IL-2 at 1 and 10 days of age, IL-10 at 1 day of age, and IL-17 during 1–24 days of age, while had a higher level of IL-6 at 24 days of age (*p* < 0.05).

The changes in ileal immuno-cytokine levels of different pig breeds during suckling and weaning periods are shown in Figure 3. The IL-1β and IL-2 levels in the TB piglets were higher (*p* < 0.05) at 21 days of age than at other days of age, and those levels were the lowest (*p* < 0.05) in the TB piglets at 1 day of age. In XB piglets, ileal IL-1β, IL-10, and IFN-γ levels at 10 and 24 days of age and the IL-6 level at 24 days of age were higher (*p* < 0.05) than those at 1 and 21 days of age, whereas the IL-17 level during 10–24 days of age and the TNF-α level at 10 days of age were higher (*p* < 0.05) than those at 1 day of age. Moreover, the IL-2 level in the XB piglets was higher (*p* < 0.05) at 10 days of age than at other days of age. In DR piglets, the IL-1β level at 1, 10, and 24 days of age, IL-6 and IL-10 levels at 1 and 24 days of age, IFN-γ level at 1 day of age, and TNF-α level at 1 and 10 days of age were lower (*p* < 0.05) than those at 21 days of age. Among different pig breeds at the same age, ileal immuno-cytokine (including IL-1β, IL-2, IL-6, IL-10, IL-17, IFN-γ, and TNF-α) levels at 1 and 21 days of age and IL-2, IL-10, IFN-γ, and TNF-α levels at 24 days of age were lower (*p* < 0.05) in the TB and XB piglets, while the IL-6 level was higher (*p* < 0.05) at 24 days of age than those in the DR piglets, and the IL-17 level in the TB piglets was lower (*p* < 0.05) at 24 days of age than that in the XB and DR piglets. Moreover, at 10 days of age, IL-1β and IL-17 levels in the DR piglets and the IL-2 level in the DR and XB piglets were higher (*p* < 0.05) compared with the TB piglets, whereas the IL-10 level was lower (*p* < 0.05) in the TB and XB piglets compared with the DR piglets. There were no significant changes (*p* > 0.05) in the IL-6, IFN-γ, and TNF-α levels among different pig breeds at 10 days of age.

### 2.5. Changes in Ileal Gene Expressions Related to Immune Function of Different Pig Breeds during Suckling and Weaning Periods

The changes in ileal gene expressions related to the immune function of different pig breeds during suckling and weaning periods are presented in Figure 4 and Figure 5. The TB piglets had up-regulated (*p* < 0.05) expression levels of *IL-10* and lipopolysaccharide-binding protein (*LBP*) at 24 days of age and *IL-6* at 10 days of age compared to other days of age. The expression levels of *IL-10* at 21 days of age, *TNF-α* at 24 days of age, and *IFN-γ* at 21 and 24 days of age were up-regulated (*p* < 0.05) in TB piglets compared to those at 1 day of age. Moreover, the expression level of *IL-17* in TB piglets was up-regulated (*p* < 0.05) at 24 days of age compared to that at 1 and 10 days of age. The XB piglets had down-regulated (*p* < 0.05) expression levels of *IL-10* at 1 day of age compared to the other days of age and *IL-1β* at 10 and 24 days of age compared to that at 1 and 21 days of age, while had up-regulated (*p* < 0.05) expression levels of *LBP* and *IFN-γ* at 24 days of age compared to those at other days of age, as well as IL-17 at 24 days of age compared to that at 10 and 21 days of age. The DR piglets had up-regulated (*p* < 0.05) expression levels of *IL-6* at 10 and 24 days of age compared to those at 1 and 21 days of age, as well as *IL-10* at 24 days of age compared to that at other days of age, while had a down-regulated (*p* < 0.05) expression level of *IL-10* at 1 day of age compared to that at other days of age.

Among different pig breeds at the same age (Figure 4 and Figure 5), the TB and XB piglets had down-regulated (*p* < 0.05) expression levels of interleukin-1 receptor-associated kinase 1 (*IRAK1*) and myeloid differentiation factor 88 (*MyD88*) at 10 and 24 days of age and the cluster of differentiation 14 (*CD14*) at 21 and 24 days of age, as well as the nuclear factor kappa B (*NF-κB*) and *IL-6* at 24 days of age, while had up-regulated (*p* < 0.05) expression levels of tumor necrosis factor receptor-associated factor 6 (*TRAF6*). The TB piglets had down-regulated (*p* < 0.05) expression levels of *IL-17* at 10 days of age than that of the DR piglets but, had an up-regulated (*p* < 0.05) expression of receptor-interact protein 2 (*RIP2*) at 24 days of age than that of the DR piglets. The expression level of *LBP* in XB piglets was up-regulated (*p* < 0.05) at 24 days of age than that in the TB and DR piglets, while the *NF-κB* at 10 and 21 days of age and *IL-1β* at 1 day of age were down-regulated (*p* < 0.05) in the TB piglets than those of the XB and DR piglets.

## 3. Discussion

The gastrointestinal health of newborn piglets is critical as it can directly influence the digestion and absorption of nutrients and the immune function of piglets [14]. Newborn piglets have an immature immune system and rely on passive protection from the sow until weaning. The passive immune level of piglets can be obtained from breast milk and reaches its peak at 7 days of age and then gradually decreases, while the autoimmune system of piglets begins to function at 4–5 weeks of age [9,15]. During the first three weeks, piglets presented the lowest immunity in their life, and their resistance to pathogens was extremely poor [16]. At this time, weaning reduces the level of circulating antibodies, inhibits cellular immunity, and then affects the growth and development of piglets [17]. However, Chinese indigenous pigs show excellent characteristics of strong stress resistance, which may be related to their immune responses due to their genetic stability and other stimulation factors. Therefore, this study compared the developmental changes in immune function and immune response to weaning stress during suckling and weaning periods in TB, XB, and DR piglets. The results showed that the developmental immune function changes in these three pig breeds were different and that Chinese indigenous pigs had a stronger immune ability to deal with weaning stress, which may be related to the regulation of the MyD88/NF-κB signaling pathway.

Liver is the key front-line immune organ, plays a crucial role in lipid metabolism and immune regulation, and its development is closely correlated with the growth of piglets [18]. A recent study indicated that the growth of piglets was associated with the liver index, and suckling piglets at 7 days of age exhibited a higher liver index than the piglets at 3 days of age [19]. In the present study, the TB piglets had a higher liver index at 10 days of age compared with the XB and DR piglets, while the XB piglets had a higher liver index at 21 days of age compared with the DR piglets, suggesting that the immune development of TB pigs began at an earlier age and that the XB pigs had stronger stress tolerance at weaning stress than the DR pigs, which may be one of the reasons for the stress resistance of Chinese indigenous pigs.

The spleen is the largest secondary lymphoid organ with extensive immune functions in the body and plays an important role in hematopoiesis and erythrocyte clearance [20]. The evidence showed that the 7-day-old piglet suffered diarrhea induced by *Clostridium perfringens* type C, and the spleen plays an immune defense against *Clostridium perfringens* infection through antigen processing and presentation of the peroxisome proliferator-activated receptor signaling pathway [21]. In the present study, the spleen index of TB and XB pigs increased with age, and there were no significant changes in the spleen index before and after weaning, suggesting that the immune function of TB and XB pigs gradually matured with age and that weaning stress did not alter the spleen immune function. Moreover, constant increases in the spleen index indicate that the spleen is growing at a constant rate compared to the growth of the pig’s body.

The change in blood leukocytes are important indicator of immune system status [22]. For instance, eosinophils are considered to be important cells involved in the regulation of the innate mucosal immune system, and they play an immune regulatory role by releasing cationic proteins stored in cytoplasmic granules [23]. Basophils are effector cells of innate immunity and play a central role in the pathogenesis of protective immunity [24]. The WBC, RBC, and PCT in the blood play an important role in immune monitoring and effector functions in lymphoid and peripheral tissues. In the disease state, due to the decreased proliferation and the inhibition of functional RBC, the body is prone to anemia and other symptoms, and the levels of MCH and MCHC in blood cell parameters are reduced [25]. In the present study, at 24 days of age, TB piglets had a higher Bas% than the XB piglets, XB piglets had a lower Eos count than that at 21 days of age, and TB and XB piglets had a lower MCHC than the DR piglets, indicating that the changes in the levels of plasma blood cells in the DR pigs were higher than those in the TB and XB pigs after weaning, while those in the TB pigs were higher than those in the XB pigs. These findings suggest that the changes in the composition of blood immune cells may result in differences in immune function among different pig breeds.

Weaning is the most important window for the adaptive immune development of piglets. Colostrum and breast milk include abundant immune and nutritional substances, such as IgA, cytokines, oligosaccharides, and proteins, which influence the health status of suckling pigs [26,27]. In order to absorb more immunoglobulin, the intestinal wall of newborn piglets forms a transient antibody penetration when there is “intestinal closure” within three days after birth [28]. In the present study, the plasma and ileal IgA, IgG, and IgM levels of the three Chinese indigenous breeds of piglets showed a downward trend from the third week. In accordance with our findings, Meishan pigs had a downward IgG level after four weeks of age [29]. We speculate that the decreased immunoglobulin levels are most likely due to the fact that the weaned piglet could not receive immunoglobulin from sow milk and had inadequate intestinal immunoglobulin secretion.

The secretory immunoglobulin obtained from breast milk after birth is crucial for piglets to resist pathogens at the early stage [30]. The IgG in sow colostrum accounts for 70–80% of the total immunoglobulin and mediates protective inflammation [2]. In the present study, XB piglets had a higher plasma IgG level during suckling and weaning periods compared with the TB and DR piglets, which indicates that XB piglets showed heterosis in obtaining maternal IgG for immune protection. Moreover, the ileal IgG level of TB and XB piglets at 1 and 21 days of age was lower than that of the DR piglets, suggesting that these piglets had higher ileal IgG circulation than the DR piglets, which may be related to intestinal permeability. The intestinal physical barrier is affected by several immune and non-immune molecules; in turn, the increased intestinal permeability may lead to the loss of immune molecules in the intestine [31]. Our previous studies have shown that the intestinal permeability of Bama mini-pigs was higher in the nursery and growing periods than the Landrace pigs, but there was no difference in the finishing stage [32]. Combined with the findings of the present study, the decrease of the ileal IgG level in TB and XB piglets may be related to the increase of IgG flow to blood circulation caused by the higher intestinal permeability.

IgM can prevent bacterial and fungal infections because it plays an important role in promoting mucosal tolerance and forming a healthy microecological environment in the intestine. In the present study, TB and XB piglets had a lower ileal IgM level during suckling and weaning periods, while XB piglets had a lower plasma IgM level at 1 day of age compared with the DR piglets, suggesting that TB and XB piglets have lower ileal IgM circulation than the DR piglets, which may be related to the conversion between IgM and sIgM and may also be related to its combination with the intestinal microbiota. In addition to being released into circulation by the spleen and plasma cells from lymph nodes and bone marrow, IgM also enters mucosal secretions as sIgM, which binds to a wider range of intestinal bacteria to form specific protective recognition [33]. The specific mechanism of ileal IgM circulation differences among different pig breeds needs to be further explored.

Moreover, IgA, as the main antibody category in mucosal surface secretions, mediates multiple protective effects and has anti-inflammatory and pro-inflammatory effects [34]. Evidence has shown that the intestinal mucosal immune system of healthy piglets produces IgA in response to antigen stimulation, in contrast to gnotobiotic pigs, where IgA production is blocked and there is little diversity [35]. The present study showed that TB and XB piglets had a higher plasma IgA level during suckling and weaning periods than the DR piglets, suggesting that these piglets received more IgA from breast milk during the suckling period. The XB piglets had a higher plasma IgA level after the weaning period than the TB and DR piglets, suggesting that XB pigs show hybridization advantages. This might be the result of the higher IgA level in breast milk, although the lactation yield of TB and XB sows may be lower than the DR sows. A previous study also showed that Hampshire and Landrace × Large white sows had higher IgA levels in colostrum than the Large white and Large white sows [36].

Cytokines play a central role in immune cell development, differentiation, and regulation. Pro-inflammatory cytokines mainly include IL-1β, IL-17, IFN-γ, and TNF-α, which are important indicators of the degree of inflammation in the host body [37]. Previous studies showed that the IFN-γ level in 14-day-old piglets increased after 7 days but decreased at 21 days of the weaning, whereas the IFN-γ level in jejunal mucosa increased after weaning [9]. In the present study, IFN-γ and IL-17 levels of DR piglets were higher during suckling and weaning periods compared to the TB and XB piglets, indicating that the DR piglets had a higher inflammatory state than the TB and XB piglets during these periods. Major anti-inflammatory cytokines include IL-6 and IL-10, which are vital immunomodulatory molecules. In the present study, the DR piglets had a lower IL-6 level at 24 days of age than the TB and XB piglets, which showed that the DR pigs have a weaker anti-inflammatory regulatory ability than the TB and XB pigs after weaning. Collectively, these findings suggest that the TB and XB piglets have stronger immune regulatory responses to weaning stress than the DR piglets.

Activation of the NF-κB signaling pathway regulates the expression of pro-inflammatory cytokines [38]. In the production of cytokines, the transcription factor of NF-κB activated by TLRs initiates the downstream MyD88 signaling pathway, and then MyD88 activates the NF-κB pathway by recruiting leukocyte-associated kinases (IRAK) and TRAF6, which not only pathways the maturation and differentiation of immune cells, but also promotes the expression of inflammatory cytokines [39]. The present study showed that ileal *IRAK1*, *CD14*, *MyD88*, *NF-κB*, and *RIP2* expression levels of DR piglets were up-regulated at 24 days of age in comparison with the TB and XB piglets. These results are in agreement with previous studies that reported that weaning stress up-regulates the expression levels of *MyD88* and *NF-κB*, which are associated with immunity and inflammation of piglets during weaning, and these expressions declined after weaning [40,41], which also coincides with the changes in immune cytokines levels in the present study. These findings suggest that the weaning stress in Chinese indigenous pigs is stronger than that in DR pigs, and this different mechanism may be related to the inhibition of the MyD88/NF-κB signaling pathway. Nevertheless, further confirmation is needed for MyD88 and NF-κB protein evaluation in order to find the correlation with cytokines and also to clarify the MyD88/NF-κB activation mechanisms.

## 4. Materials and Methods

### 4.1. Animals and Study Design

This study was conducted at the Institute of Subtropical Agriculture, Chinese Academy of Sciences, Changsha, Hunan, China. A total of 30 litters of newborn piglets (ten litters per breed), including TB, XB, and DR piglets, were selected from sows with similar parities (2–3) and litter size (9–11), respectively. Chinese indigenous piglets (TB and XB piglets) were obtained from Xiangcun High-Tech Agricultural Co., Ltd. (Loudi, China), and DR piglets were obtained from Tianxin Breeding Share Co., Ltd. (Changsha, China). Suckling piglets were not given creep feed during lactation. After weaning at 21 days of age, piglets were fed with a creep feed. The other feeding management followed commercial feeding management protocols. During this trial, piglets did not receive any vaccinations.

### 4.2. Sample Collection

At 1, 10, 21 (weaned), and 24 (3 days after weaning) days of age, 12 h after the last feeding, ten piglets (male: female, 1:1) per breed (one piglet from each litter), close to the average body weight of the litter, were selected for sampling. The average body weights of TB, XB, and DR piglets were 1.50 ± 0.25, 1.39 ± 0.19, and 1.82 ± 0.43 kg at 1 day of age, 2.65 ± 0.78, 2.63 ± 0.87, and 3.86 ± 0.98 kg at 10 days of age, 5.06 ± 1.12, 3.42 ± 1.12, and 6.01 ± 1.91 kg at 21 days of age, and 4.37 ± 1.78, 3.22 ± 0.99, and 5.74 ± 1.59 kg at 24 days of age, respectively. Blood samples (5 mL) were collected from the anterior vena cava into potassium ethylenediamine tetraacetate (K_2_-EDTA) anticoagulant tubes (Aosaite, Shangdong, China) for hematological parameters analysis. In addition, 10 mL blood samples from each pig were also drawn into heparin sodium anticoagulant tubes (Aosaite, Shangdong, China) to obtain plasma by centrifuging at 4 °C and 3500× *g* for 15 min for immunoglobulin and immuno-cytokine analysis. The pigs were euthanized for sampling after intramuscular injection of Zoletil^®^ 50 (Beijing Lab Anim Tech Develop Co., Ltd., Beijing, China). A 5-cm section of the posterior segment of the ileal tissue was collected, frozen in liquid nitrogen, and stored at −80 °C to determine the immune function-related indexes and gene expressions of related signal molecules. The liver and spleen were isolated and weighed.

### 4.3. Analysis of Visceral Organ Index

The recorded weights of the liver and spleen were used to calculate the visceral organ index according to the following formula:Visceral organ index (g/kg) = organ weight (g)/body weight (kg)

### 4.4. Analysis of Blood Hematological Parameters

The blood hematological parameters, including WBC, Neu, Neu%, Lym, Lym%, monocyte count (Mon), Mon%, eosinophil count (Eos), Eos%, Bas, Bas%, RBC, HGB, HCT, MCV, RDW-CV, RDW-SD, MCH, MCHC, PLT, MPV, PDW, and PCT were measured using the BC-5000VET automatic blood cell analyzer (Shenzhen Mindray Biomedical Electronics Co., Ltd., Shenzhen, China) within 2 h after blood collection.

### 4.5. Analysis of Plasma and Ileal Immunoglobulin and Immuno-Cytokine Contents

The plasma and ileal immunoglobulin and immuno-cytokine contents were determined using commercially available enzyme-linked immunosorbent assay (ELISA) kits for swine (Shanghai Kexing Trading Co., Ltd., Shanghai, China), following the manufacturer’s instructions. The absorbance (OD value) was measured at 450 nm using a microplate reader (Infinite M200 PRO, TECAN, Männedorf, Switzerland). The concentration of total protein in ileal tissues was detected with the BCA protein assay kit (Beyotime, Shanghai, China), and the final contents of immunoglobulin and immuno-cytokine of ileal tissues were normalized to the total protein in each sample.

### 4.6. Analysis of Immune-Related Gene Expressions in the Ileum

The total RNA from ileum tissue samples was extracted using Trizol reagent (Accurate Biology, Hunan, China). The concentration of the extracted RNA was measured using a NanoDrop 2000 (Thermo Fisher Scientific, Waltham, MA, USA), and the purity of the RNA solution was determined using the A_260_/A_280_ ratio ranges from 1.8 to 2.1. The RNA quality was determined by agarose gel electrophoresis. The reverse transcriptional program was performed at 37 °C for 15 min and 95 °C for 5 s according to the manufacturer’s protocol (Accurate Biology). The primers used in this study are listed in Appendix A. β-actin was used as the housekeeper gene for gene expression normalization. Real-time PCR was performed with SYBR^®^ Green Premix *Pro Taq* HS qPCR Kit (Accurate Biology). The PCR cycling conditions were as follows: initial denaturation at 94 °C for 30 s, then denaturation at 94 °C for 5 s and annealing at 55 °C for 30 s with 40 cycles, and a final extension at 72 °C for 30 s. The cycling was performed on the LightCycler^®^ 480 II Real-Time PCR System (Roche, Basel, Switzerland). The method of comparing the Ct value was used to calculate the relative expression of the target gene in comparison with β-actin, and the 2^−ΔΔCt^ value was used to represent the relative gene expression level.

### 4.7. Statistical Analysis

All data in this study are expressed as means ± standard error of the mean (SEM). The data were analyzed using the SPSS 22.0 software package (SPSS, Inc., Chicago, IL, USA). The differences between the means of the experimental groups were analyzed using a one-way analysis of variance and Tukey’s multiple comparison test. Probability values of <0.05 were considered to indicate statistical significance, and probability values between 0.05 and 0.10 were considered a trend.

## 5. Conclusions

In summary, there are differences in immune development and immune response to weaning stress among different breeds of pigs. The differences in visceral organ development between Chinese indigenous piglets and DR piglets may gradually decline with age. The TB and XB piglets had a higher plasma IgA level during suckling and weaning periods than the DR piglets, and XB piglets show hybridization advantages in obtaining plasma IgG. Meanwhile, the TB and XB piglets had lower ileal IFN-γ and IL-17 levels but higher ileal anti-inflammatory cytokine levels (such as IL-6 and IL-10) at weaning than the DR piglets, which may be related to the MyD88/NF-κB signaling pathway mediated immune regulation. These findings will provide a physio-biochemical basis for explaining the stress resistance of Chinese indigenous pigs and are of great significance for the protection and utilization of indigenous pig resources.

## Figures and Tables

**Figure 1 ijms-23-15781-f001:**
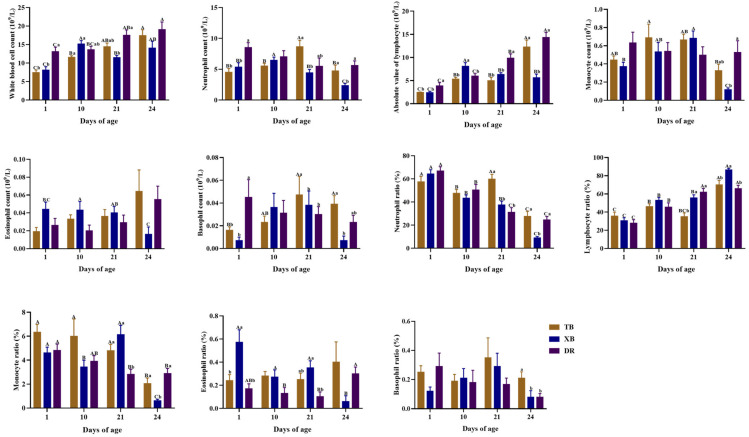
Changes in blood leukocyte composition of different pig breeds during suckling and weaning periods. Data are presented as means with their pooled SEM (*n* = 10). Different capital letters (A, B, and C) indicate significant differences between ages in the same breed (*p* < 0.05), and different small letters (a and b) indicate significant differences between breeds at the same age (*p* < 0.05). DR, Duroc piglet; TB, Taoyuan black piglet; XB, Xiangcun black piglet.

**Figure 2 ijms-23-15781-f002:**
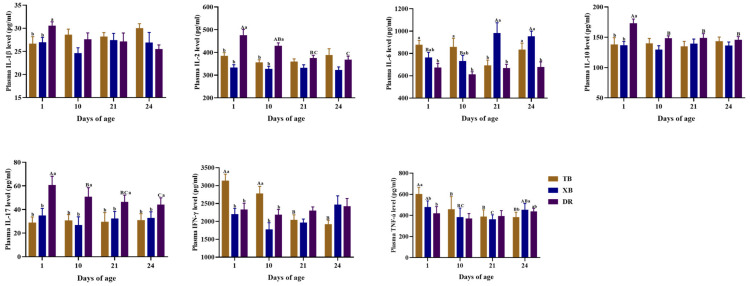
Changes in plasma immuno-cytokine levels of different pig breeds during suckling and weaning periods. Data are presented as means with their pooled SEM (*n* = 10). Different capital letters (A, B, and C) indicate significant differences between ages in the same breed (*p* < 0.05), and different small letters (a and b) indicate significant differences between breeds at the same day of age (*p* < 0.05). IL, interleukin; IFN, interferon; TNF, tumor necrosis factor; DR, Duroc piglet; TB, Taoyuan black piglet; XB, Xiangcun black piglet.

**Figure 3 ijms-23-15781-f003:**
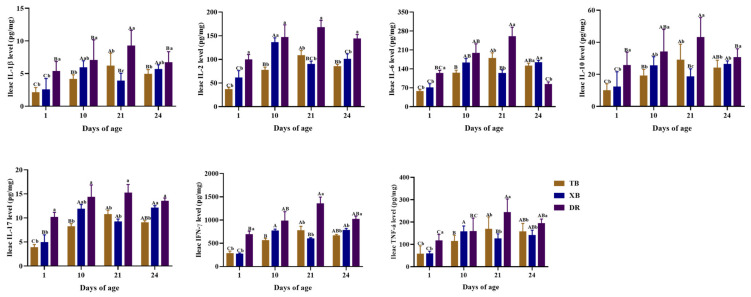
Changes in ileal immuno-cytokine contents of different pig breeds during suckling and weaning periods. Data are presented as means with their pooled SEM (*n* = 10). Different capital letters (A, B, and C) indicate significant differences between ages in the same breed (*p* < 0.05), and different small letters (a, b, and c) indicate significant differences between breeds at the same age (*p* < 0.05). IL, interleukin; IFN, interferon; TNF, tumor necrosis factor; DR, Duroc piglet; TB, Taoyuan black piglet; XB, Xiangcun black piglet.

**Figure 4 ijms-23-15781-f004:**
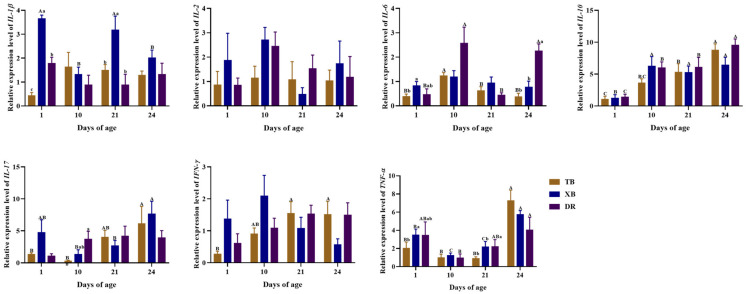
Changes in gene expression levels of ileal immuno-cytokines of different pig breeds during suckling and weaning periods. Data are presented as means with their pooled SEM (*n* = 10). Different capital letters (A, B, and C) indicate significant differences between ages in the same breed (*p* < 0.05), and different small letters (a, b and c) indicate significant differences between breeds at the same day of age (*p* < 0.05). IL, interleukin; IFN-γ, interferon γ; TNF-α, tumor necrosis factor-α; DR, Duroc piglet; TB, Taoyuan black piglet; XB, Xiangcun black piglet.

**Figure 5 ijms-23-15781-f005:**
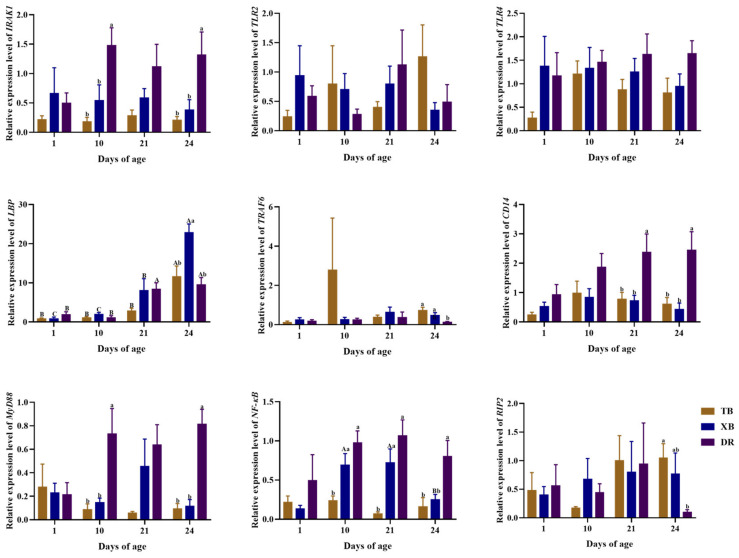
Changes in expression levels of ileal genes related to TLR-MyD88-NF-κB signaling pathway of different pig breeds during suckling and weaning periods. Data are presented as means with their pooled SEM (*n* = 10). Different capital letters (A, B, and C) indicate significant differences between ages in the same breed (*p* < 0.05), and different small letters (a and b) indicate significant differences between breeds at the same day of age (*p* < 0.05). IRAK1, interleukin-1 receptor-associated kinase 1; LBP, lipopolysaccharide binding protein; TRAF6, tumor necrosis factor receptor-associated factor 6; CD14, cluster of differentiation 14; MyD88, myeloid differentiation factor 88; NF-κB, nuclear factor kappa B; RIP2, receptor-interact protein 2; DR, Duroc piglet; TB, Taoyuan black piglet; XB, Xiangcun black piglet.

**Table 1 ijms-23-15781-t001:** Changes in organ indexes of different pig breeds during suckling and weaning periods.

Items	TB	XB	DR	SEM	*p* Values
Liver index (g/kg)					
1 day of age	21.62 ^C^	25.83	32.20	3.42	0.108
10 days of age	31.21 ^Aa^	27.41 ^b^	25.90 ^b^	0.84	<0.001
21 days of age	25.98 ^Bab^	27.37 ^a^	23.96 ^b^	0.89	0.039
24 days of age	25.64 ^B^	26.59	26.85	0.92	0.625
SEM	0.81	0.97	2.99		
*p* values	<0.001	0.622	0.261		
Spleen index (g/kg)					
1 day of age	1.24 ^B^	1.55 ^B^	2.28	0.38	0.337
10 days of age	2.10 ^A^	2.17 ^A^	1.95	0.15	0.567
21 days of age	2.04 ^A^	2.18 ^A^	2.08	0.20	0.872
24 days of age	1.87 ^A^	1.75 ^AB^	1.93	0.13	0.610
SEM	0.12	0.18	0.33		
*p* values	<0.001	0.035	0.868		

Data are presented as means with their pooled SEM (*n* = 10). Different capital letters (A, B, and C) in the same column indicate significant differences among different days of age (*p* < 0.05), and different small letters (a and b) in the same row indicate significant differences among different pig breeds (*p* < 0.05). DR, Duroc piglet; TB, Taoyuan black piglet; XB, Xiangcun black piglet.

**Table 2 ijms-23-15781-t002:** Changes in plasma immunoglobulin levels of different pig breeds during suckling and weaning periods.

Items (μg/mL)	TB	XB	DR	SEM	*p* Values
IgA					
1 day of age	25.56 ^a^	25.88 ^ABa^	17.59 ^Bb^	1.43	<0.001
10 days of age	24.30 ^a^	22.93 ^Ba^	18.84 ^Bb^	1.03	0.002
21 days of age	24.71 ^a^	27.31 ^Aa^	17.56 ^Bb^	1.05	<0.001
24 days of age	23.75 ^b^	28.05 ^Aa^	23.00 ^Ab^	1.37	0.031
SEM	1.05	1.19	1.44		
*p* values	0.667	0.022	0.035		
IgG					
1 day of age	323.29 ^a^	207.00 ^Bb^	340.81 ^a^	17.11	<0.001
10 days of age	322.48 ^a^	221.64 ^Bb^	322.61 ^a^	9.95	<0.001
21 days of age	311.81 ^a^	206.66 ^Bb^	333.64 ^a^	10.30	<0.001
24 days of age	324.95 ^a^	270.43 ^Ab^	321.69 ^a^	15.71	0.035
SEM	9.71	16.85	13.43		
*p* values	0.769	0.035	0.706		
IgM					
1 day of age	33.33 ^Aab^	29.32 ^b^	34.41 ^Aa^	1.43	0.043
10 days of age	32.00 ^AB^	28.63	30.03 ^B^	1.25	0.179
21 days of age	28.26 ^B^	31.61	29.26 ^B^	1.06	0.090
24 days of age	28.66 ^B^	30.55	29.07 ^B^	1.13	0.473
SEM	1.37	1.30	0.97		
*p* values	0.030	0.390	0.001		

Data are presented as means with their pooled SEM (*n* = 10). Different capital letters (A and B) in the same column indicate significant differences among different days of age (*p* < 0.05), and different small letters (a and b) in the same row indicate significant differences among different pig breeds (*p* < 0.05). Ig, immunoglobulin; DR, Duroc piglet; TB, Taoyuan black piglet; XB, Xiangcun black piglet.

**Table 3 ijms-23-15781-t003:** Changes in ileal immunoglobulin levels of different pig breeds during suckling and weaning periods.

Items (μg/mg)	TB	XB	DR	SEM	*p* Values
IgA					
1 day of age	1.88 ^Cb^	2.91 ^Bab^	3.88 ^Ba^	0.45	0.014
10 days of age	4.39 ^B^	5.73 ^A^	6.88 ^A^	0.97	0.212
21 days of age	6.08 ^A^	5.19 ^A^	6.67 ^A^	0.67	0.298
24 days of age	4.89 ^ABb^	6.30 ^Aab^	8.13 ^Aa^	0.64	0.005
SEM	0.56	0.59	0.92		
*p* values	<0.001	0.001	0.018		
IgG					
1 day of age	19.42 ^Cb^	25.64 ^Cb^	65.13 ^a^	7.81	0.001
10 days of age	46.65 ^B^	66.57 ^A^	69.93	9.81	0.212
21 days of age	61.37 ^ABb^	46.81 ^Bb^	98.91 ^a^	8.36	0.001
24 days of age	68.42 ^A^	59.14 ^AB^	88.10	8.10	0.051
SEM	6.94	6.01	11.63		
*p* values	<0.001	<0.001	0.158		
IgM					
1 day of age	1.74 ^Cb^	1.91 ^Cb^	5.81 ^Ba^	0.51	<0.001
10 days of age	3.94 ^Bb^	5.54 ^Ab^	8.27 ^ABa^	0.89	0.007
21 days of age	5.67 ^Ab^	4.14 ^Bb^	11.48 ^Aa^	1.04	<0.001
24 days of age	4.94 ^ABc^	6.14 ^Ab^	7.80 ^Ba^	0.38	<0.001
SEM	0.49	0.37	1.15		
*p* values	<0.001	<0.001	0.013		
sIgA					
1 day of age	1.41 ^Cb^	1.82 ^Cb^	5.01 ^a^	0.45	<0.001
10 days of age	2.52 ^Bb^	5.45 ^Aa^	5.19 ^a^	0.57	0.002
21 days of age	3.41 ^Ab^	3.73 ^Bb^	6.98 ^a^	0.57	<0.001
24 days of age	4.10 ^Ab^	5.87 ^Aa^	4.73 ^ab^	0.41	0.016
SEM	0.30	0.43	0.69		
*p* values	<0.001	<0.001	0.109		

Data are presented as means with their pooled SEM (*n* = 10). Different capital letters (A, B, and C) in the same column indicate significant differences among different days of age (*p* < 0.05), and different small letters (a, b, and c) in the same row indicate significant differences among different pig breeds (*p* < 0.05). Ig, immunoglobulin; sIgA, secretory immunoglobulin A; DR, Duroc piglet; TB, Taoyuan black piglet; XB, Xiangcun black piglet.

## Data Availability

The data presented in the study are included in the article. Further inquiries can be directed to the corresponding authors.

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
