# Peer review of "Developmental Changes of Immunity and Different Responses to Weaning Stress of Chinese Indigenous Piglets and Duroc Piglets during Suckling and Weaning Periods"

_ijms, 2022, doi:10.3390/ijms232415781_

Round 1
Reviewer 1 Report
The research by Ding et al. provides developmental changes in the immunity and different responses to the weaning stress of piglets of different pig breeds during suckling and weaning periods. The authors evaluated organ index, hematological parameters, immune cytokines contents of plasma and intestine, and gene expressions related to immune function and compared those parameters among different pig breeds and ages.
Overall, the research paper is interesting, well-designed, easy to follow, and uses appropriate methodologies to achieve its objectives. The findings of this paper will enrich the knowledge for further understanding the immune system development of pigs during suckling and weaning and will also provide the resources for indigenous pig production.
However, several issues need to be considered/improved before resubmission for acceptance in the International Journal of Molecular Sciences. Specific comments are as follows:
Comments:
L 33: gastrointestinal tract.
L 36: two-fold.
L 58-59: What types of gene expressions? It is not clear from the current statement. Authors need to modify/rephrase it for better clarification.
L 64-66: Which adverse reactions? Needs more details.
L 69-73: The authors only described the characteristics of TB. Authors should add a brief description of the XB pigs' characteristics.
L 88-89: delete this sentence.
L 101-114: Abbreviations should be defined at their first appearance in the text; then use throughout the text.
L 107-108: From Fig. 1J; it is lower than the XB pigs; Also, lettering is not consistent. Authors should cross-check this.
L 122-125: MCH of TB piglets at 21 and 24 days was not comparable at 10 days of age; because the superscript "AB" share the superscript "B". Authors should cross-check the results.
L 144-145: IgM in TB piglets at 21 and 24 days of age was not significantly lower than at 10 days of age; cross-check the results.
L 147-149: IgA in XB piglets at 10 days of age was lower than that at 21 and 24 days of age.
L 157-158: This sentence should be moved in L 152.
L 171-172: L 171-172: IgA level at 1 and 24 days, only significantly lower in the TB piglets (not in the XB piglets; shared superscript "a" with DR piglets) that the DR piglets. There were a lot more such errors in this paper. Authors should cross-check the results description carefully.
L 194-195: There is no statistical comparison of TNF-alpha level at 10 days of age among different pig breeds at the same age.
L 215-216: IL-2 in TB piglets is not higher at 10 days of age than that at other days of age. Cross-check the results.
L 219-223: The results description is not clear/correct according to the Figure 3. Authors need to cross-check carefully.
L 242: TNF-alpha at 21 days of age in TB piglets was not up-regulated than 1 day of age.
L 243: IFN-gamma at 24 days of age is not comparable with 10 days of age.
L 245-246: IL-17 was up-regulated at 24 days of age.
L 248: No lettering for XB piglets (Figure 4F).
L 236-270: The results descriptions are not consistent with the figures.
L 427-429: What was the euthanization procedure for sampling? What was the length of the tissues for sampling?
L 434-445: Abbreviations of these hematological parameters should be defined at their first appearance in the text.
Figures:
(1) Figure legends T, X, and D should be changed as TB, XB, and DR.
(2) It not necessary to indicate each figure with different letters (A, B, C, etc.). If the authors want to keep these lettering; then it’s better to cite every subsection of the figure in the text (e.g., Figure 1A, Figure 1B, Figure 1C, etc.).
Author Response
Reviewer 1
The research by Ding et al. provides developmental changes in the immunity and different responses to the weaning stress of piglets of different pig breeds during suckling and weaning periods. The authors evaluated organ index, hematological parameters, immune cytokines contents of plasma and intestine, and gene expressions related to immune function and compared those parameters among different pig breeds and ages.
Overall, the research paper is interesting, well-designed, easy to follow, and uses appropriate methodologies to achieve its objectives. The findings of this paper will enrich the knowledge for further understanding the immune system development of pigs during suckling and weaning and will also provide the resources for indigenous pig production.
However, several issues need to be considered/improved before resubmission for acceptance in the International Journal of Molecular Sciences. Specific comments are as follows:
Sincerest thanks for your valuable comments on our manuscript. Please be noted that all the changes to the manuscript are indicated in red color.
Comments:
- L 33: gastrointestinal tract.
Response: We have revised in L 41.
- L 36: two-fold.
Response: We have revised this issue in the manuscript (L 45).
- L 58-59: What types of gene expressions? It is not clear from the current statement. Authors need to modify/rephrase it for better clarification.
Response: We have revised this issue, as follows:
Line 65-69: Furthermore, weaning also leads to the impaired immune function of the small intestine of piglets, resulting in a decline in lymphocyte immune function, increases in the intestinal pro- and anti-inflammatory cytokines and related gene expressions, and changes in intestinal morphology.
- L 64-66: Which adverse reactions? Needs more details.
Response: We added more details in the revised manuscript.
Line 74-76: Chinese indigenous pigs have shown good adaptability to the specific feeding environment and feed resources in intensive production, and have the characteristics of resistance to adverse reactions (i.e., pre- and post-weaning stress, roughage feeding, etc.).
- L 69-73: The authors only described the characteristics of TB. Authors should add a brief description of the XB pigs' characteristics.
Response: We added more details about Xiangcun black pigs in the revised manuscript.
Line 80-83: Xiangcun black (XB) pigs, a cross-bred of TB pigs (as the female parent) and Duroc (DR) pigs (as the male parent), have the characteristics of strong stress resistance, large litter size, faster growth and development, high feed utilization rate, and excellent meat quality.
- L 88-89: delete this sentence.
Response: We have deleted this sentence.
- L 101-114: Abbreviations should be defined at their first appearance in the text; then use throughout the text.
Response: We have revised this issue throughout the text.
- L 107-108: From Fig. 1J; it is lower than the XB pigs; Also, lettering is not consistent. Authors should cross-check this.
Response: We have revised this issue in the revised manuscript; also, checked the results carefully.
Line 116-119: Among different pig breeds at the same age, the neutrophil count (Neu) and basophil count (Bas) levels and Neu% at 21 days of age and basophil ratio (Bas%) at 24 days of age were higher (P < 0.05) in TB piglets compared with those of in the XB and DR piglets.
- L 122-125: MCH of TB piglets at 21 and 24 days was not comparable at 10 days of age; because the superscript "AB" share the superscript "B". Authors should cross-check the results.
Response: Thanks for your valuable suggestions. We have revised this issue, as following:
Line 127-132: In TB piglets, the hemoglobin (HGB) and hematocrit (HCT) levels were higher (P < 0.05) at 24 days of age than those at 1, 10, and 21 days of age, whereas the coefficient variation of red blood cell volume distribution width (RDW-CV) and standard deviation in red cell distribution width (RDW-SD) levels were lower (P < 0.05) at 1 and 24 days of age than those at 10 and 21 days of age.
- L 144-145: IgM in TB piglets at 21 and 24 days of age was not significantly lower than at 10 days of age; cross-check the results.
Response: Thanks for your correction. We have revised this issue, as following:
Line 147-148: The IgM level in TB piglets was lower (P < 0.05) at 21 and 24 days of age than that at 1 day of age.
- L 147-149: IgA in XB piglets at 10 days of age was lower than that at 21 and 24 days of age.
Response: Thanks for your correction. We have revised this issue, as following:
Line 150-153: In XB piglets, the IgG level was higher (P < 0.05) at 24 days of age than that at other days of age, and the IgA level was lower (P < 0.05) at 10 days of age than that at 21 and 24 days of age, while there were no changes (P > 0.05) in IgA and IgM levels at different days of age.
- L 157-158: This sentence should be moved in L 152.
Response: Thanks for your suggestion. We have revised this issue in the manuscript.
- L 171-172: L 171-172: IgA level at 1 and 24 days, only significantly lower in the TB piglets (not in the XB piglets; shared superscript "a" with DR piglets) that the DR piglets. There were a lot more such errors in this paper. Authors should cross-check the results description carefully.
Response: We appreciate the reviewer’s concerns. We have checked the results description carefully throughout the text.
Line 176-177: the IgA level in the TB piglets was lower (P < 0.05) at 1 and 24 days of age than that in the DR piglets.
- L 194-195: There is no statistical comparison of TNF-alpha level at 10 days of age among different pig breeds at the same age.
Response: We have revised this issue, as following:
Line 185-188: In TB piglets, the interferon (IFN)-γ level was higher (P < 0.05) at 1 and 10 days of age than that at 21 and 24 days of age, as well as the tumor necrosis factor (TNF)-α level at 1 day of age than that of other days of age.
- L 215-216: IL-2 in TB piglets is not higher at 10 days of age than that at other days of age. Cross-check the results.
Response: We have revised this issue, as following:
Line 203-205: The IL-1β and IL-2 levels in TB piglets were higher (P < 0.05) at 21 days of age than those at three days of age, and those levels were the lowest in TB piglets (P < 0.05) at 1 day of age.
- L 219-223: The results description is not clear/correct according to the Figure 3. Authors need to cross-check carefully.
Response: We have revised this issue, as following:
Line 213-216: ileal immuno-cytokines (including IL-1β, IL-2, IL-6, IL-10, IL-17, IFN-γ, and TNF-α) levels at 1 and 21 days of age and IL-2, IL-10, IFN-γ, and TNF-α levels at 24 days of age were lower in the TB and XB piglets.
- L 242: TNF-alpha at 21 days of age in TB piglets was not up-regulated than 1 day of age.
Response: We have revised this issue, as following:
Line 227-230: The TB piglets had up-regulated (P < 0.05) expression levels of IL-10 and lipopolysaccharide-binding protein (LBP) at 24 days of age and IL-6 at 10 days of age than those at other days of age.
- L 243: IFN-gamma at 24 days of age is not comparable with 10 days of age.
Response: We have revised this issue, as following:
Line 230-232: The expression levels of IL-10 at 21 days of age, TNF-α at 24 days of age, and IFN-γ at 21 and 24 days of age were up-regulated (P < 0.05) in TB piglets than those at 1 day of age.
- L 245-246: IL-17 was up-regulated at 24 days of age.
Response: We have revised this issue, as following:
Line 234-238: The XB piglets had down-regulated (P < 0.05) expression levels of IL-10 at 1 day of age than that at other three days of age, and the IL-1β at 10 and 24 days of age than that at 1 and 21 days of age, while had up-regulated (P < 0.05) expression levels of LBP and IFN-γ at 24 days of age than those at other days of age and the IL-17 at 24 days of age (P < 0.05) than that at 10 and 21 days of age.
- L 248: No lettering for XB piglets (Figure 4F).
Response: We have revised this issue, as following:
Line 238-242: The DR piglets had up-regulated (P < 0.05) expression levels of IL-6 at 10 and 24 days of age than that at 1 and 21 days of age, and IL-10 at 24 days of age than that at other three days of age, while had down-regulated (P < 0.05) expression level of IL-10 at 1 day of age than that at other days of age.
- L 236-270: The results descriptions are not consistent with the figures.
Response: We have revised this issue, as following:
Line 243-254: Among different pig breeds at the same age showed in Figures 4 and 5, the TB and XB piglets had down-regulated (P < 0.05) expression levels of interleukin-1 receptor-associated kinase 1 (IRAK1) and myeloid differentiation factor 88 (MyD88) at 10 and 24 days of age and CD14 at 21 and 24 days of age, as well as the nuclear factor kappa B (NF-kB) and IL-6 at 24 days of age, while had up-regulated (P < 0.05) expression levels of tumor necrosis factor receptor-associated factor 6 (TRAF6). The TB piglets had down-regulated (P < 0.05) expression levels of IL-17 at 10 days of age than that of DR piglets, while up-regulated expression of and receptor-interact protein 2 (RIP2) at 24 days of age than that of DR piglets. The expression level of LBP in XB piglets was up-regulated (P < 0.05) at 24 days of age than that in the TB and DR piglets, while the NF-kB at 10 and 21 days of age and IL-1β at 1 day of age were down-regulated (P < 0.05) in TB piglets than those of XB and DR piglets.
- L 427-429: What was the euthanization procedure for sampling? What was the length of the tissues for sampling?
Response: We have revised this issue, as following:
Line 426-431: The pigs were euthanized for sampling after intramuscular injection of Zoletil@ 50 (Beijing Lab Anim Tech Develop Co., Ltd., Beijing, China). A 5-cm section of the posterior segments of the ileal tissue was collected, frozen in liquid nitrogen, and stored at -80°C to determine the immune function-related indexes and gene expressions of related signal molecules. The liver and spleen were isolated and weighed.
- L 434-445: Abbreviations of these hematological parameters should be defined at their first appearance in the text.
Response: We have defined all abbreviations at their first appearance in the text.
Figures:
(1) Figure legends T, X, and D should be changed as TB, XB, and DR.
Response: We have replaced “T, X, and D” with “TB, XB, and DR” in figure legends.
(2) It not necessary to indicate each figure with different letters (A, B, C, etc.). If the authors want to keep these lettering; then it’s better to cite every subsection of the figure in the text (e.g., Figure 1A, Figure 1B, Figure 1C, etc.).
Response: We have modified all figures.
Reviewer 2 Report
Dear Editor
The aim of the study entitled ijms-2069178-‘’Developmental Changes of Immunity and Different Responses to Weaning Stress of Chinese Indigenous Piglets and Duroc Piglets during Suckling and Weaning Periods‘.
The present study deals with an interesting topic and fits with the scope of the journal.
The development of the immune system of pigs during the pre-and post-natal periods is one of major importance for a healthy pig. Generally, the current article is a well-designed and well-written work, providing new information for immunity and stress at the weaning stage.
Therefore, I suggest that the article be accepted for publication, under major revision.
Please let me congratulate you on the quality of your journal and thank you for giving me the opportunity to contribute as a reviewer.
Comments for the authors
Major comments
- Please explain why you selected piglets only from sows with parity 2-3 and not from primiparous sows (parity 1)
- Provide details about the vaccination program of sows. Was the vaccination program the same for all sows? The vaccination program is an important factor in the study of immunity in weaning piglets
- Did selected piglets receive any vaccination during this trial?? If yes, did all piglets receive the same vaccination scheme??
- Did all litters derive from sows of the same farm or different farms? Please give more details about the farm or farms.
- Please explain why the suckling piglets did not receive creep feed. Give details about the feeding of suckling piglets: did they receive milk replacers??
- Provide more information about the selected piglets: mean+sd BW, number of male/female per litter, castrated male or barrows…
Minor comments
§ L66-70: add appropriate references
§ L83: .. during the suckling and weaning ..
§ L93: Table 1- correct the uppercase letters (small letters a, b...)
§ L410: change the subtitle (e.g., Animals and study design)
§ L472: add appropriate references
Author Response
Reviewer 2
The aim of the study entitled ijms-2069178-‘Developmental Changes of Immunity and Different Responses to Weaning Stress of Chinese Indigenous Piglets and Duroc Piglets during Suckling and Weaning Periods‘.
The present study deals with an interesting topic and fits with the scope of the journal.
The development of the immune system of pigs during the pre-and post-natal periods is one of major importance for a healthy pig. Generally, the current article is a well-designed and well-written work, providing new information for immunity and stress at the weaning stage. Therefore, I suggest that the article be accepted for publication, under major revision.
Please let me congratulate you on the quality of your journal and thank you for giving me the opportunity to contribute as a reviewer.
Sincerest thanks for your response and reviewer comments on our manuscript. Please be noted that all the changes to the manuscript are indicated in red color.
Comments for the authors
Major comments
- Please explain why you selected piglets only from sows with parity 2-3 and not from primiparous sows (parity 1)
Response: This is because the survival rate of piglets in the first litter of sows was low, and the immune system of sows currently was not mature and susceptible to pathogen infection, while the reproductive performance and immune system of sows in the 2-3 litter are more stable. Therefore, in order to select healthy piglets with similar body weight for subsequent research experiments, piglets from sows with 2-3 parities were used for this study.
- Provide details about the vaccination program of sows. Was the vaccination program the same for all sows? The vaccination program is an important factor in the study of immunity in weaning piglets
Response: Chinese indigenous sows and Duroc sows were kept separately by professional commercial companies, and their vaccine procedures were subjected to the commercial companies’ protocols.
- Did selected piglets receive any vaccination during this trial?? If yes, did all piglets receive the same vaccination scheme??
Response: The selected piglets were not vaccinated during the experimental period.
- Did all litters derive from sows of the same farm or different farms? Please give more details about the farm or farms.
Response: Chinese indigenous sows were kept in Xiangcun High-Tech Agriculture Co., Ltd. (Loudi, Hunan, China), Duroc sows were kept in Hunan Tianxin Breeding Share Co., Ltd. (Changsha, Hunan, China).
- Please explain why the suckling piglets did not receive creep feed. Give details about the feeding of suckling piglets: did they receive milk replacers??
Response: Piglets received only sow's breast milk during lactation. After weaning, the piglets were separated from the sows and began to take creep feed, as follows:
Line 409-412: Suckling piglets were not given creep feed during lactation. After weaning at 21 days of age, piglets were fed with a creep feed. The other feeding management followed the commercial feeding management protocols. During this trial, piglets have not received any vaccination.
- Provide more information about the selected piglets: mean+sd BW, number of male/female per litter, castrated male or barrows…
Response: We added more information about the selected piglets in the revised manuscript.
L 414-420: At 1, 10, 21 (weaned), and 24 (3 days after weaning) days of age, 12 h after the last feeding, ten piglets (male: female, 1:1) per breed (one piglet from each litter), close to the average body weight of litter, were selected for sampling. The average body weight of TB, XB, and DR piglets were 1.50 ± 0.25, 1.39 ± 0.19, and 1.82 ± 0.43 kg at 1 day of age, 2.65 ± 0.78, 2.63 ± 0.87, and 3.86 ± 0.98 at 10 days of age, 5.06 ± 1.12, 3.42 ± 1.12, and 6.01 ± 1.91 kg at 21 days of age, and 4.37 ± 1.78, 3.22 ± 0.99, and 5.74 ± 1.59 kg at 24 days of age, respectively.
Minor comments
- L66-70: add appropriate references
Response: We had added references in L 66-70, as following:
Line 65-69: Furthermore, weaning also leads to the impaired immune function of the intestinal mucosa of piglets, resulting in a decline in lymphocyte immune function, significant increases in the pro- and anti-inflammatory cytokines and gene expressions in intestinal tissue, and changes in intestinal morphology [9,10].
- Cao, S.; Hou, L.; Sun, L.; Gao, J.; Gao, K.; Yang, X.; Jiang, Z.; Wang, L. Intestinal morphology and immune profiles are altered in piglets by early-weaning. Int. Immunopharmacol. 2022, 105, 1567ï€1570.
- Wang, L.; Yan, S.; Li, J.; Li, Y.; Ding, X.; Yin, J.; Xiong, X.; Yin, Y.; Yang, H. Rapid communication: The relationship of enterocyte proliferation with intestinal morphology and nutrient digestibility in weaning piglets. J. Anim. Sci. 2019, 97, 353ï€358.
- L83: during the suckling and weaning.
Response: We have revised this issue, as following:
Line 86-89: we hypothesized that understanding the underlying mechanism of developmental changes in the intestinal immune function of indigenous pig breeds will shed new light into nutrition approaches to improve their health during suckling and weaning periods.
- L93: Table 1- correct the uppercase letters (small letters a, b...)
Response: We have revised this issue in Table 1.
- L410: change the subtitle (e.g., Animals and study design)
Response: We have revised this issue in Line 402.
- L472: add appropriate references
Response: This is the conclusion of this study without reference.
Round 2
Reviewer 2 Report
I accept the response of authors to my comments. I have no more suggestions and comments. I suggest that this article could be published.